Senescent epithelial cells remodel the microenvironment for the progression of oral submucous fibrosis through secreting TGF-β1

Wang Zijia 1
Han Ying 1
Peng Ying 1
Shao Shuhui 1
Nie Huanquan 1
Xia Kun 2
Xiong Haofeng xionghaofeng@sklmg.edu.cn 1 3 4 5
Su Tong sutong@csu.edu.cn 1 3 4 5
1 Department of Oral and Maxillofacial Surgery, Center of Stomatology, Xiangya Hospital, Central South University , Changsha , Hunan , China
2 Center for Medical Genetics & Hunan Key Laboratory of Medical Genetics, School of Life Sciences, Central South University , Changsha , Hunan , China
3 Research Center of Oral and Maxillofacial Tumor, Xiangya Hospital, Central South University , Changsha , China
4 Institute of Oral Precancerous Lesions, Central South University , Changsha , China
5 National Clinical Research Center for Geriatric Disorders, Xiangya Hospital , Changsha , China
Rokaya Dinesh
Electronic publication date: 2023 Apr 19
Publication date: 2023
Volume: 11
Electronic Location ID: e15158
Received 2022 Dec 19; Accepted 2023 Mar 13
Copyright: ©2023 Wang et al.
Copyright year: 2023
Copyright holder: Wang et al.
License: This is an open access article distributed under the terms of the Creative Commons Attribution License, which permits unrestricted use, distribution, reproduction and adaptation in any medium and for any purpose provided that it is properly attributed. For attribution, the original author(s), title, publication source (PeerJ) and either DOI or URL of the article must be cited.
License URL: https://creativecommons.org/licenses/by/4.0/

Keywords: Oral submucous fibrosis, Epithelial cell, Cellular senescence, Senescence-associated secretory phenotype, Transforming growth factor β

Funding: The National Natural Science Foundation 81873717 82170973 The Clinical Research Fund of the National Clinical Research Center for Geriatric Disorders of Xiangya Hospital 2021LNJJ08 This work was supported by the National Natural Science Foundation (81873717 and 82170973) and the Clinical Research Fund of the National Clinical Research Center for Geriatric Disorders of Xiangya Hospital (2021LNJJ08). The funders had no role in study design, data collection and analysis, decision to publish, or preparation of the manuscript.

==============================
Objectives

Cellular senescence is strongly associated with fibrosis and tumorigenesis. However, whether the epithelium of oral submucous fibrosis (OSF) undergoes premature senescence remains unclear. This study investigates the roles of senescent epithelial cells in OSF.

Methods

The immunohistochemistry and Sudan black B staining were performed to identify epithelium senescence in OSF tissues. Arecoline was used to induce human oral keratinocytes (HOKs) senescence. The cell morphology, senescence-associated β galactosidase activity, cell counting Kit 8, immunofluorescence, quantitative real-time PCR, and western blot assay were used to identification of senescent HOKs. The enzyme-linked immunosorbent assay was exerted to evaluate the levels of transforming growth factor β1 (TGF-β1) in the supernatants of HOKs treated with or without arecoline.

Results

The senescence-associated markers, p16 and p21, were overexpressed in OSF epithelium. These expressions were correlated with alpha-smooth actin (α-SMA) positively and proliferating cell nuclear antigen (PCNA) negatively. Moreover, Sudan black staining showed that there was more lipofuscin in OSF epithelium. In vitro, HOKs treated with arecoline showed senescence-associated characteristics including enlarged and flattened morphology, senescence-associated β galactosidase staining, cell growth arrest, γH2A.X foci, upregulation of p53, p21, and TGF-β1 protein levels. Moreover, senescent HOKs secreted more TGF-β1.

Conclusions

Senescent epithelial cells are involved in OSF progression and may become a promising target for OSF treatment.

Introduction

Oral submucous fibrosis (OSF) is a premalignant mucosal disease with inflammation and progressive fibrosis. Although OSF is a multifactorial disease, epidemiological studies indicated that betel nut chewing is one of the most significant risk factors for OSF (Angadi & Rekha, 2011; Tilakaratne et al., 2006). During betel nut chewing, its physical and chemical stimuli directly impose on oral mucosa. Among these adverse stimuli, arecoline is a recognized chemical pathogenic factor for OSF, mainly activating fibroblasts to promote the collagen deposition and suppress the collagen degradation (Yang et al., 2021). As fibrosis progresses, it can severely impair oral functions, such as eating, speaking, and swallowing. Moreover, 1.5–15% of OSF may eventually develop into oral squamous cell carcinoma (OSCC) (Shih et al., 2019). Currently, the treatment of OSF can only relieve its symptoms, but not cured it (Shih et al., 2019). Therefore, it is urgent to find a potent therapy for OSF to reverse or even block its progression and malignant transformation.

Numerous studies have shown that cellular senescence is closely related to fibrosis and cancer (Huang et al., 2021; Yao et al., 2021). Senescent cells exhibit cell cycle arrest and develop a secretory profile composed of growth factors, inflammatory cytokines, chemokines, and other bioactive molecules, termed the senescence-associated secretory phenotype (SASP) (Shmulevich & Krizhanovsky, 2021). Cell cycle arrest prevents damaged cells from proliferating, and the secretion of SASP recruits immune cells to eliminate senescent cells, making cellular senescence a critical protective mechanism (Herranz & Gil, 2018). Senescent cells, however, have a detrimental effect on neighboring cells by generating excessive SASP when they abnormally accumulate in tissues (Loo et al., 2020). For example, senescent alveolar type II cells contributed to lung fibrogenesis by activating the profibrotic phenotype of alveolar macrophages through secreting SASP (Rana et al., 2020). Through SASP, senescent hepatic stellate cells enhanced hepatocellular carcinoma cell growth and accelerated xenograft tumor growth (Li et al., 2020). Besides, senotherapy, which targets senescent cells, has been demonstrated that is an effective treatment for fibrosis and cancer (Amor et al., 2020). However, the mechanism of cellular senescence in OSF is still poorly understood.

Up to date, there has been relatively little study on cellular senescence in OSF. Rehman et al. (2016) reported in 2016 that senescent fibroblasts remodel the tissue microenvironment to promote the progression of OSF to OSCC through releasing SASP. However, as the first barrier, far too little attention has been paid to the roles of the epithelium in the pathogenesis of OSF. Since epithelial atrophy is a characteristic pathological manifestation of OSF and OSCC originates from the epithelium, we speculated that epithelial cells of OSF may undergo premature senescence, which participates in OSF progression.

TGF-β1 is a multifunctional cytokine that plays a role in numerous physiological and pathological processes (Kim, Sheppard & Chapman, 2018). In OSF, TGF-β1 is a major profibrotic factor, mainly by promoting fibroblast proliferation and transforming into myofibroblasts to produce extracellular matrix (Hsieh et al., 2018). Moreover, TGF-β1 may also play an important role in the tumorigenesis of OSF, as it has been shown to regulate epithelial-mesenchymal transition, immunosuppression, and stemness (Cave et al., 2020; Yeh et al., 2018). Gao, Ling & Wu (1997) reported in 1997 that the mRNA level of TGF-β1 was the highest expressed in OSF epithelial tissue compared to the healthy mucosal tissue and oral lichen planus. In addition, our previous study also confirmed that TGF-β1 was predominantly expressed in OSF epithelium, which was significantly higher than that in healthy mucosal epithelium (Wang et al., 2018). These studies suggested that OSF epithelium is a major source of TGF-β1, which may be involved in the fibrotic progression and carcinomagenesis of OSF by secreting excessive TGF-β1. However, it remains unclear what pathological changes occur in OSF epithelium that causes the excessive secretion of TGF-β1. We hope this study will contribute to the exploration of targeted therapy for OSF.

In this study, we demonstrated that OSF epithelial cells undergo premature senescence and could play roles in OSF progression.

Materials and Methods

Patient samples

A total of 30 samples were collected from the center of stomatology, Xiangya Hospital from September 2019 to May 2020. 21 OSF tissues were obtained from OSF patients without OSF therapy. Nine healthy oral mucosa (NOM) samples were obtained from healthy volunteers during surgical removal of the lower third molars. All samples were diagnosed by pathological examination based on the 2005 World Health Organization classification system. All research subjects gave written informed consent in accordance with the Helsinki Declaration. This study was approved by the Medical Ethics Committee of Xiangya Hospital, Central South University (Ethical Application Ref: 201910837).

Immunohistochemistry assay

The immunohistochemistry was performed as described previously (Min et al., 2020). Tissues were fixed with 4% paraformaldehyde for 24 h, and then were dehydrated in graded alcohol solutions and embedded in paraffin. The paraffin-embedded tissues were sliced into 5-µm sections. The following primary antibodies were used in immunohistochemistry: p16 (1:500, ab108349; Abcam, Cambridge, UK), p21 (1:50, 2947T; Cell Signaling Technology, Danvers, MA, USA), alpha-smooth actin (α-SMA) (1:200, ab5694; Abcam), proliferating cell nuclear antigen (PCNA) (1:16000, 2586S; Cell Signaling Technology). Phosphate-buffered saline (PBS) was used instead of the primary antibody as a negative control, and NOM tissues were used as a positive control. All slides were observed by microscopy (Leica Microsystems, Cambridge, UK). The staining results were quantitatively evaluated using the Image-Pro-Plus 5.0 software to measure the mean optical density (MOD) of positive p16, p21, PCNA, and α-SMA in OSF tissues (MOD = integral optical density (IOD)/area of interest) (Hu et al., 2021). The epithelium or subepithelial areas of NOM and OSF tissues were marked as the interest areas for analysis.

Sudan black B staining

To detect lipofuscin accumulation in OSF epithelium, Sudan black B solution was used to stain the lipofuscin. Sudan black B staining (S109070; Aladdin) was carried out as described previously (Georgakopoulou et al., 2013).

Cell culture

Human Oral Keratinocytes (HOKs) is currently the only normal oral epithelial cell line available, which were cultured in complete Alpha MEM (10% FBS and 1% penicillin-streptomycin added) at 37°, 5% CO2 incubator (Wang et al., 2018). As the major components of betel nut, we used arecoline to induce HOKs senescence. The half-maximal inhibitory concentrations (IC50) of arecoline in HOKs were 200 µg/ml within 24 h and 160 µg/ml within 48 h by CCK8 assay, and then arecoline concentrations were grouped accordingly. HOKs were randomly divided into six groups, and after starvation in serum-free medium overnight, they were incubated with complete medium containing different concentrations of arecoline (0, 20, 40, 80, 160, and 320 µg/ml) for 24 h. Then, all groups were replaced with complete medium, and the medium was changed every 3 days for 10–14 days.

Senescence-associated β galactosidase (SA-β-gal) activity assay

The SA-β-gal activity was performed using a SA-β-gal staining Kit (CS0030; Sigma-Aldrich, St. Louis, MO, USA) following the manufacturer’s instructions. Briefly, after senescence induction, HOKs were seeded into 24-well culture plates at a density of 1 ×104 cells per well. The next day, the cells were washed three times with PBS and fixed by cell fixative solution in the SA-β-gal staining Kit for 7 min at room temperature. Then, the cells were washed three times with PBS to remove the solution. The staining mixture containing X-gal prepared following the instruction was incubated with the cells in a regular incubator at 37° overnight without CO2. The cells were observed by microscopy (Leica Microsystems, Cambridge, UK). A total of 5 fields in each group were selected under the microscope and the total cells and SA-β-gal positive cells in each field were counted by ImageJ 5.0. The proportion of SA-β-gal positive cells was obtained by calculating the ratio of the average SA-β-gal positive cell number to the average total cell number in each group.

Immunofluorescence assay

Immunofluorescence was applied to examine the expression of phosphor-histone H2A.X (γH2A.X) in HOKs treated with different concentrations of arecoline. After senescence induction, the cells were seeded into 24-wells culture plates at 1 ×104 per well density. The next day, the cells were fixed with 4% paraformaldehyde for 10 min. Subsequently, the cells were permeabilized with 0.1% Triton X-100 solution and blocked with 10% goat serum (ZLI-9056, ZSGB-BIO). After PBS washing, the cells were incubated with monoclonal rabbit anti- γH2A.X primary antibody (1:400, 9718, Cell Signaling Technology) at 4° overnight. The next day, the cells were incubated with cy3-conjugated secondary antibody (AP132C, Sigma) for 1 h in dark, and then counterstained with 4′,6-Diamidino-2-phenylindole (C1002, Beyotime) at room temperature for 1 min. Immunofluorescence images were acquired using a fluorescence microscope (Leica Microsystems, Cambridge, UK). A total of five fields in each group were selected under the microscope and the total cells and γH2A.X positive cells in each field were counted by ImageJ 5.0. The proportion of γH2A.X positive cells was obtained by calculating the ratio of the average γH2A.X positive cell number to the average total cell number in each group.

Cell proliferation assay

Cell proliferation ability was assessed by Cell Counting Kit 8 (CCK8) (A311-01; Vazyme). CCK8 assay was performed by the manufacturer’s protocol. After senescence induction, HOKs were seeded into 96-well plates at a density of 5000 cells per well and continually cultured for 0, 24, 48, 72, and 96 h. Then, CCK8 solution (10 µl) was added to each well for 2 h incubation at 37°. The plates were detected at the indicated time (0, 24, 48, 72, and 96 h) using a microplate reader (Synergy 2; Biotek, Winooski, VT, USA) at a wavelength of 450 nm.

Quantitative reverse transcription polymerase chain reaction (qRT-PCR) assay

Total RNA was extracted from HOKs treated with different concentrations of arecoline using TRIzol reagent (15596026; Invitrogen, Waltham, MA, USA). The total RNA was reverse transcribed to cDNA using RevertAid™ Master Mix, with DNase I Kit (M16325; Thermo Scientific, Waltham, MA, USA) according to the manufacturer’s protocol. qPCR was performed using Maxima SYBR Green qPCR Master Mix (K0252; Thermo Scientific, Waltham, MA, USA) on a Real-Time PCR Detection System (CFX96; Bio-Rad, Hercules, CA, USA) according to the manufacturer’s protocol. All qPCR primers were listed in Table 1. The relative mRNA expression was calculated using the 2−ΔΔCt method. GAPDH served as housekeeping gene. The value of negative control group was used to calculate the relative fold change in expression value.

Western blotting assay

The total protein was extracted from HOKs treated with different concentrations of arecoline using SDS lysis buffer (P0013G; Beyotime). The total protein concentration was detected using Pierce™ BCA Protein Assay Kit (23225; Thermo Fisher Scientific). A total of 40 µg protein was added to 10% polyacrylamide gel and transferred to polyvinylidene fluoride (PVDF) membranes. The membranes were blocked with 5% skim milk for 1 h at room temperature. After PBST washing three times, the membranes were incubated with primary antibodies overnight at 4° overnight: p16 (1:2000, ab108349, Abcam), p21 (1:2000, 2946; Cell Signaling Technology), p53 (1:1000, 48818; Cell Signaling Technology), transforming growth factorβ1 (TGF-β1) (1:1000, ab215715; Abcam), and GAPDH (1:1000, 5174T, Cell Signaling Technology). Then, the membranes were incubated with the corresponding horseradish peroxidase-conjugated secondary antibodies (anti-Rabbit/mouse 1:5000∼10000, SA00001-1/2; Proteintech) for 1 h at room temperature. The membranes were developed using SuperSignal™ West Pico PLUS chemiluminescent substrate (34580; Thermo Fisher Scientific) and detected by a chemiluminescent imaging system (MiniChemi 610; Sage Creation, Beijing, China). and analyzed using the ImageJ 5.0 version software.

Enzyme-linked immunosorbent (ELISA) assay

Levels of TGF-β1 in supernatants were measured by the Human TGF-β1 Elisa Kit (PT880; Beyotime, Jiangsu, China) following the manufacturer’s protocol. The absorbance of samples was detected at 450 nm using a microplate reader (Synergy 2; Biotek, Winooski, VT, USA). The TGF-β1 concentration of each sample was determined based on a standard curve we prepared according to the manufacturer’s protocol.

Table 1 Table-revision.

Primer sequences used in this study.

Name	Sequence	Tm value	
p16	F 5′-GGGTTTTCGTGGTTCACATCC-3′	55.10	
R 5′-CTAGACGCTGGCTCCTCAGTA-3′	60.77	
p21	F 5′-TGTCCGTCAGAACCCATGC-3′	56.22	
R 5′-AAAGTCGAAGTTCCATCGCTC-3′	57.38	
p53	F 5′-CAGCACATGACGGAGGTTGT-3′	57.6	
R 5′-TCATCCAAATACTCCACACGC-3′	55.79	
TGF-β1	F 5′-CTAATGGTGGAAACCCACAACG-3′	56.50	
R 5′-TATCGCCAGGAATTGTTGCTG-3′	57.77	
GAPDH	F 5′-CCATGGGTGGAATCATATTGGA-3′	57.60	
R 5′-TCAACGGATTTGGTCGTATTGG-3′	58.11	

Statistical analysis

All experiments were conducted in triplicate and repeated at least three times. All data were shown as mean ± standard deviation. Continuous variables were assessed by the student’s t test or Mann–Whitney U test according to the normality test. Spearman’s correlation was used to analyze the correlation. P value <0.05 was considered statistically significant. Statistical analysis was performed using SPSS 24.0 software.

Results

The epithelium of OSF undergoes premature senescence

p16 and p21 belong to cyclin-dependent kinase inhibitors, which are major drivers of the cycle arrest in senescence and have been confirmed to be one of the defining features of senescent cells (Calcinotto et al., 2019). Therefore, we detected the expressions of p16 and p21 in the epithelium of OSF and healthy tissues. We found that p16 and p21 were overexpressed in OSF epithelium compared with the healthy epithelium (Figs. 1A–1D). In contrast, PCNA, which indicates cellular proliferation, was highly expressed in the epithelium of NOM (Figs. 1A, 1B, 1E). Moreover, the expressions of p16 and p21 were negatively correlated with PCNA in OSF epithelial tissues (Figs. 1H–1K).

Figure 1 The expressions and correlation of p16, p21, PCNA, and α-SMA in normal oral mucosa (NOM) and oral submucous fibrosis (OSF).

(A–B) The representative images of p16, p21, PCNA, and α-SMA expressions detected by immunohistochemistry (IHC) in NOM and OSF tissues. (C–F) Quantification of the expressions of p16, p21, PCNA, and α-SMA in NOM and OSF tissues. (G) The representative images of lipofuscin expression detected by Sudan Black B staining in the epithelial of NOM and OSF tissues (black arrow). (H) The correlation between p16 and α-SMA expressions in NOM and OSF tissues. (I) The correlation between p16 and PCNA expressions in NOM and OSF tissues. (J) The correlation between p21 and α-SMA expressions in NOM and OSF tissues. (K) The correlation between p21 and PCNA expressions in NOM and OSF tissues. * p < 0.05, ** p < 0.01, *** p < 0.001, and **** p < 0.0001. Scale bar: 100 µm in 100 ×, 25 µm in 400 ×.

To further confirm epithelial cells senescence in OSF, we detected the accumulation of lipofuscin in OSF epithelium by Sudan Black staining. Lipofuscin is abnormally accumulated in senescent cells, another well-established senescence marker (Georgakopoulou et al., 2013). After Sudan black B staining, we found that the lipofuscin was widely distributed in OSF epithelium (Fig. 1G).

Correlation between the epithelial cell senescence and myofibroblast activation in human OSF tissues

To further investigate the correlation between the epithelial cell senescence and myofibroblast activation in OSF tissue, we detected the expression of α-SMA that is a characteristic marker of myofibroblast activation. We observed that there were α-SMA positive spindle cells located in the lamina propria adjacent to the epithelium of OSF (Fig. 1B). In sharp contrast, we only detected rings of α-SMA positive cells representing blood vessels in the lamina propria of healthy tissues (Fig. 1A). We hardly found the α-SMA positive spindle cells in healthy tissues. Moreover, the expression of α-SMA in OSF tissues was extremely higher than that in healthy tissues (p <0.001, Fig. 1F). Further analysis revealed that α-SMA expression in OSF tissues was positively correlated with the expressions of senescence markers, p21 and p16 (p <0.01, Fig. 1H).

Arecoline induced senescence-like cell morphology and suppressed cell proliferation in HOKs

Since OSF is an oral mucosal disease associated with the chewing of betel nut, we choose arecoline to induce HOKs senescence due to it is a major component of betel nut. To confirm HOKs developing senescence, we examined senescence-associated phenotypes in HOKs.

After HOKs were treated with different concentrations of arecoline for 24 h, they gradually appeared an enlarged and flattened morphology, following another 5–7 days of culture (Fig. 2A). Meanwhile, the enlarged and flattened HOKs also showed SA-β-gal positive staining (Fig. 2A). The proportion of SA-β-gal positive HOKs was gradually increased with arecoline concentration. Among different concentrations of arecoline, the 320 µg/ml group had the highest proportion of SA-β-gal positive HOKs compared with other groups (p <0.05, Fig. 2B). Moreover, the surface area of SA-β-gal positive cells was about 10-fold more than that of SA-β-gal negative cells (p <0.05, Fig. 2C). CCK8 assay showed that compared with that of other groups, the proliferation ability of the 320 µg/ml group was largely suppressed after senescence induction (p <0.05, Fig. 2D).

Figure 2 The performance of senescence-associated beta galactosidase (SA-β-gal) staining, cell morphology, and cell proliferation in human oral keratinocytes (HOKs) after senescence induction.

(A) The representative images of SA-β-gal staining and cell morphology in HOKs treated with different concentrations of arecoline (SA-β-gal positive cell: black arrow, green). (B) The proportion of SA-β-gal positive cells in HOKs treated with different concentrations of arecoline. (C) Quantification of the surface area of SA-β-gal negative HOKs and SA-β-gal positive HOKs (SA-β-gal negative HOK: black arrow, SA-β-gal positive HOK: white arrow). (D) The cell proliferation of HOKs treated with different concentrations of arecoline. * p < 0.05, ** p < 0.01, *** p < 0.001, and **** p < 0.0001. Scale bar: 100 µm in 100 ×.

Arecoline induced DNA damage response (DDR) in HOKs

The phosphorylated-histone H2A.X was detected in HOKs after senescence induction, a hallmark of DNA double-strand breakage in cells, because DNA damage stimulates DNA damage response (DDR) that triggers senescence (Di Micco et al., 2021; Gorgoulis et al., 2019). We found that γH2A.X was mainly detected in HOKs treated with high concentrations of arecoline (160 and 320 µg/ml) (Fig. 3A). The proportion of γH2A.X positive HOKs was elevated with arecoline concentration, and that in the 320 µg/ml group was significantly higher than in other groups (p <0.05, Fig. 3B). Moreover, we observed that γH2A.X was mainly expressed in the nucleus of the enlarged and flatted HOKs (Fig. 3C).

Figure 3 The expression of phosphor-histone H2A.X (γ H2A.X) in HOKs treated with different concentrations of arecoline after senescence induction.

(A) The representative images of γ H2A. X expression in HOKs treated with the different concentrations of arecoline (blue: DAPI, red: γ H2A.X, γ H2A.X positive cell: white arrow). (B) The proportion of γ H2A.X positive cells in HOKs treated with differentiation concentrations of arecoline. (C) The representative images of the senescent HOK (blue: DAPI, red: γ H2A.X). * p < 0.05, ** p < 0.01, *** p < 0.001, and **** p < 0.0001. Scale bar: 100 µm in 100 ×, 50 µm in 200 ×.

Arecoline-induced HOK senescence was associated with p53/p21 pathway

Cell cycle arrest of senescent cells is controlled by activation of the p53/p21 or p16/Rb pathway (Gorgoulis et al., 2019). Since most of HOKs in the 320 µg/ml group undergo senescence, we compared the levels of p16, p21, and p53 between the groups of the negative control (0 µg/ml) and the 320 µg/ml. We found that p21 expression was dramatically upregulated at the protein and mRNA level (p < 0.01, Figs. 4A, 4B), while p53 expression was on the rise only at the protein level in the 320 µg/ml group (p <0.05, Figs. 4A, 4C). No difference was observed in p16 at protein and mRNA levels in both groups (Fig. 4A).

Figure 4 The expressions of p16, p21, p53, and TGF-β1 in HOKs treated with 0 µg/ml arecoline or 320 µg/ml arecoline after senescence induction.

(A) The representative images of p16, p21, p53, TGF-β1, and GAPDH protein levels in HOKs treated with 0 µg/ml or 320 µg/ml arecoline. (B) Quantification expressions of p21 at protein and mRNA levels in HOKs treated with 0 µg/ml or 320 µg/ml arecoline. (C) Quantification expressions of p53 at protein and mRNA levels in HOKs treated with 0 µg/ml or 320 µg/ml arecoline. (D) Quantification expressions of TGF-β1 at protein and mRNA levels in HOKs treated with 0 µg/ml or 320 µg/ml arecoline. (E) TGF-β1 expression in supernatants of HOKs treated with 0 µg/ml or 320 µg/ml arecoline. * p < 0.05, ** p < 0.01, *** p < 0.001, and **** p < 0.0001.

The synthesis and secretion of TGF-β1 are increased in senescent HOKs

Growth factors are a prominent constituent of SASP. Given that TGF-β1 is a well-known profibrotic factor, we examined the protein and mRNA levels of TGF-β1 between the groups of the negative control (0 µg/ml) and the 320 µg/ml. The protein and mRNA levels of TGF-β1 in the 320 µg/ml group were all significantly upregulated, compared with the negative control group (p <0.05, Fig. 4D). In addition, although the density of HOKs in the 320 µg/ml group after senescence induction was approximately 30–40% of that in the negative control group, we found that the levels of TGF-β1 in the supernatant of the 320 µg/ml group were increased approximately 2-fold compared to the control group (p <0.01, Fig. 4E).

Discussion

The treatments of OSF are currently used to effectively control symptoms and improve oral functions and the life quality of OSF patients (Shih et al., 2019). Yet these treatments can neither completely cure OSF nor block its transformation to OSCC. Recently, senescent cells have been shown to participate in the development of fibrotic diseases and tumors (Huang et al., 2021; Yao et al., 2021), and were regarded as a novel target for effectively suppressing the disease progression (Amor et al., 2020). In OSF, the cellular senescence of fibroblasts has received the first attention. It has been confirmed that fibroblasts can undergo senescence under adverse stimuli and participate in the process of fibrosis and malignant transformation of OSF (Pitiyage et al., 2011; Rehman et al., 2016). However, as a direct target of adverse stimuli, OSF epithelium has not received much attention. Currently, there was a speculation that the cause of epithelial atrophy in OSF may be epithelial cell senescence, whereas the senescence escape will make them susceptible to malignant transformation (Sharma et al., 2021). To our knowledge, our work is the first to focus on cellular senescence in the epithelium of OSF patients.

In the present study, we found that lipofuscin was abnormally increased in OSF epithelium. Additionally, the expressions of p16 and p21 were elevated in the epithelium of OSF by immunohistochemistry assay. Their expressions were negatively correlated to PCNA expression, which was hardly expressed in the epithelium of OSF. Both p16 and p21 are well-established markers of cellular senescence and belong to the Cyclin-Dependent Kinase Inhibitor family, mediating cell cycle arrest to suppress cell proliferation (Gorgoulis et al., 2019). These results provided evidence that OSF epithelium undergoes cellular senescence, which results in epithelial atrophy. Meanwhile, we also found that p16 and p21 expression were positively correlated with α-SMA expression, a marker of the activated myofibroblast. Activated myofibroblasts are the major source of collagen in fibrosis (Lee et al., 2021). These results indicated that the epithelial senescence was associated with the myofibroblast activation. Tian et al. (2019) have confirmed that senescent alveolar epithelial cells activated myofibroblasts through intercellular communication in lung fibrosis. These findings implied that senescent epithelial cells could activate myofibroblasts by intercellular communication. Therefore, to further investigate the roles of senescent epithelial cells, we used arecoline to induce HOKs senescence in vitro, as HOKs are immortalized epithelial cells obtained from the basal layer of oral mucosa tissues.

Arecoline has been shown to generate reactive oxygen species (ROS) in different types of cells (Shih et al., 2020; Shih et al., 2021), and ROS is one of the common triggers of cellular senescence (Di Micco et al., 2021). We found that HOKs exhibited various senescence-associated phenotypes under the short-duration stimulation of arecoline. The proportion of these cells showed a dose-dependent relationship with arecoline. A possible explanation for this might be that arecoline raised intracellular reactive oxygen species (ROS) levels in a dose-dependent manner (Yen et al., 2011). Moreover, γH2A.X foci localized in the nucleus of senescent-like cells, and the expressions of p53 and p21 were obviously higher in HOKs treated with 320 µg/ml arecoline than in HOKs without arecoline. The formation of γH2A.X foci represents the activation of DNA damage response (DDR) and dysregulation of the cell cycle (Hernandez-Segura, Nehme & Demaria, 2018). The DDR can activate the p53/p21 pathway, and p21 is responsible for initiating senescence (Di Micco et al., 2021). These findings presented that arecoline can induce HOKs senescence, which is consistent with the study of Patil et al. (2021). However, our study also provided an insight that arecoline-induced senescence in HOKs is closely linked to the p53/p21 pathway. In addition, TGF-β1 expressions were significantly increased at protein and mRNA levels in HOKs treated with 320 µg/ml arecoline, compared with HOKs without arecoline. These results indicated that senescent HOKs can synthesize more TGF-β1. Although TGF-β1 is one of the common components of SASP, the composition of SASP depends on cell type and stimulus. Subsequently, we examined TGF-β1 levels in the supernatant to determine that senescent HOKs can secrete excessive TGF-β1.

As identified in ELISA experiments, we found that secreted TGF-β1 levels in senescent HOK supernatants were approximately 2-fold higher than in negative control samples. These results suggested that senescent HOKs could release excessive TGF-β1. Our previous study reported that TGF-β1 was overexpressed in the epithelium of OSF (Wang et al., 2018). Consequently, senescent epithelial cells could be one of the major cellular sources of upregulated TGF-β1 expression in the epithelium of OSF samples.

As a pleiotropic cytokine, firstly, TGF-β1 has been demonstrated to induce myofibroblasts differentiation to facilitate fibrosis (Huang et al., 2020; Little et al., 2020; Zhang et al., 2021), and secondly, TGF-β1 can participate in tumorigenesis by mediating epithelial-mesenchymal transition, immunosuppression, and stemness (Cave et al., 2020; Yeh et al., 2018). Moreover, TGF-β1 has also been shown to have a pro-senescence effect, which can induce myofibroblasts senescence through autophagy deregulation or TGF-β -dependent mechanisms (Hassona et al., 2013; Tan et al., 2021). Hence, it raises the possibility that there may be a paracrine positive-feedback loop to amplify senescence and reinforce TGF-β1 signaling in OSF epithelium (Fig. 5).

Figure 5 The mechanism diagram for the role of senescent epithelial cells in the development of OSF.

Accordingly, targeting the senescent epithelial cells may constitute a promising therapeutic approach for OSF. Amor et al. (2020) recently reported that chimeric antigen receptor T cells can effectively eliminate senescent cells, which extend the lifespan of mice harboring lung adenocarcinoma and restore tissue homeostasis in mice with liver fibrosis.

In summary, we found that the epithelium of OSF undergoes premature senescence, and there is a positive correlation between senescent epithelial cells and myofibroblasts activation. In addition, our study indicated that senescent HOKs secrete excessive TGF-β1, which suggested that senescent epithelial cells could affect surrounding cells by TGF-β1 secretion. Moreover, this research has an important implication for developing targeted therapeutics for senescent epithelial cells in OSF, which may be beneficial for reversing and blocking its progression and malignant transformation.

Conclusions

Our study first revealed that cellular senescence occurs in OSF epithelium, and senescent epithelial cells are an important source of TGF-β1 in OSF epithelium. These results provide a new perspective for OSF target therapy against senescent epithelium.

Supplemental Information

Supplemental Information 1 Area of cell

Click here for additional data file.

Supplemental Information 2 IHC data

Click here for additional data file.

Supplemental Information 3 CCK8 data

Click here for additional data file.

Supplemental Information 4 The proportion of SA-β-gal positive cells

Click here for additional data file.

Supplemental Information 5 Data of WB, Elisa, and qRT-PCR

Click here for additional data file.

Supplemental Information 6 GAPDH for p21

Total protein (35 µg) was extracted from the negative control HOKs (lane 1, 3, 5) and senescent HOKs (lane 2, 4, 6) (lanes were ordered from left to right). GAPDH (35kDa).

Click here for additional data file.

Supplemental Information 7 The membrane of GAPDH for p21

Total protein (35 µg) was extracted from the negative control HOKs (lane 1, 3, 5) and senescent HOKs (lane 2, 4, 6) (lanes were ordered from left to right). GAPDH (35 kDa).

Click here for additional data file.

Supplemental Information 8 WB-p21

Total protein (35 µg) was extracted from the negative control HOKs (lane 1, 3, 5) and senescent HOKs (lane 2, 4, 6) (lanes were ordered from left to right). p21 (21 kDa).

Click here for additional data file.

Supplemental Information 9 The membrane of p21

Total protein (35 µg) was extracted from the negative control HOKs (lane 1, 3, 5) and senescent HOKs (lane 2, 4, 6) (lanes were ordered from left to right). p21 (21 kDa).

Click here for additional data file.

Supplemental Information 10 WB-TGF- β1

Total protein (35 µg) was extracted from the negative control HOKs (lane 1, 3) and senescent HOKs (lane 2) (lanes were ordered from left to right). TGF- β1 (44 kDa).

Click here for additional data file.

Supplemental Information 11 The membrane of TGF- β1

Total protein (35 µg) was extracted from the negative control HOKs (lane 1, 3) and senescent HOKs (lane 2) (lanes were ordered from left to right). TGF- β1 (44 kDa).

Click here for additional data file.

Supplemental Information 12 GAPDH for TGF- β1

Total protein (35 µg) was extracted from the negative control HOKs (lane 1, 3) and senescent HOKs (lane 2) (lanes were ordered from left to right). GAPDH (35 kDa).

Click here for additional data file.

Supplemental Information 13 The membrane of GAPDH for TGF- β1

Total protein (35 µg) was extracted from the negative control HOKs (lane 1, 3) and senescent HOKs (lane 2) (lanes were ordered from left to right). GAPDH (35 kDa).

Click here for additional data file.

Supplemental Information 14 GAPDH for TGF- β1

Total protein (35 µg) was extracted from the negative control HOKs (lane 1, 3, 4) and senescent HOKs (lane 2, 5) (lanes were ordered from left to right). GAPDH (35 kDa).

Click here for additional data file.

Supplemental Information 15 WB-GAPDH-p53

Total protein (35 µg) was extracted from the negative control HOKs (lane 1, 3, 5) and senescent HOKs (lane 2, 4, 6) (lanes were ordered from left to right). GAPDH (35 kDa).

Click here for additional data file.

Supplemental Information 16 The membrane of GAPDH for TGF- β1

Total protein (35 µg) was extracted from the negative control HOKs (lane 1, 3, 4) and senescent HOKs (lane 2, 5) (lanes were ordered from left to right). GAPDH (35 kDa).

Click here for additional data file.

Supplemental Information 17 WB-p53

Total protein (35 µg) was extracted from the negative control HOKs (lane 1, 3, 5) and senescent HOKs (lane 2, 4, 6) (lanes were ordered from left to right). p53 (53 kDa).

Click here for additional data file.

Supplemental Information 18 The membrane of p53

Total protein (35 µg) was extracted from the negative control HOKs (lane 1, 3, 5) and senescent HOKs (lane 2, 4, 6) (lanes were ordered from left to right). p53 (53 kDa).

Click here for additional data file.

Supplemental Information 19 WB-TGF- β1

Total protein (35 µg) was extracted from the negative control HOKs (lane 1, 3, 4) and senescent HOKs (lane 2, 5) (lanes were ordered from left to right). TGF- β1 (44 kDa).

Click here for additional data file.

We thank Professor Sun Zhijun in Hospital of Stomatology Wuhan University for supporting HOKs.

Additional Information and Declarations

Competing Interests

Author Contributions

Human Ethics

Data Availability

The authors declare there are no competing interests.

Zijia Wang conceived and designed the experiments, performed the experiments, analyzed the data, prepared figures and/or tables, authored or reviewed drafts of the article, and approved the final draft.

Ying Han performed the experiments, prepared figures and/or tables, and approved the final draft.

Ying Peng performed the experiments, prepared figures and/or tables, and approved the final draft.

Shuhui Shao performed the experiments, prepared figures and/or tables, and approved the final draft.

Huanquan Nie performed the experiments, prepared figures and/or tables, and approved the final draft.

Kun Xia conceived and designed the experiments, authored or reviewed drafts of the article, and approved the final draft.

Haofeng Xiong conceived and designed the experiments, performed the experiments, authored or reviewed drafts of the article, and approved the final draft.

Tong Su conceived and designed the experiments, authored or reviewed drafts of the article, and approved the final draft.

The following information was supplied relating to ethical approvals (i.e., approving body and any reference numbers):

This study was approved by the Medical Ethics Committee of Xiangya Hospital, Central South University (Ethical Application Ref: 201910837).

The following information was supplied regarding data availability:

The raw measurements are available in the Supplemental Files.

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
