# Peer review of "Senescent epithelial cells remodel the microenvironment for the progression of oral submucous fibrosis through secreting TGF-β1"

_PeerJ, doi:10.7717/peerj.15158_

## Round 0.1 · original submission · Minor Revisions

Based on the Reviewers comments, please correct the manuscript.

Reviewer 1 ·

Basic reporting

In your introduction, please expound further on the relation of OSF to TGF- and the related literature on its correlation with each other/ or if there is any more possible relation between the two. I see that it was explained more in the “Discussion” part, but it may be helpful for the reader to have an overview.

Regarding the research objective, something is missing on why this area should be explored. More examples should be stated for better understanding.

The role of arecoline has been explained very well in the discussion, and the narrative was straightforward.

Experimental design

The research question should also include the need to find TGF-beta in OSF. Or the possibility of finding it since it is associated with OSF.

Methods have been very well described.

Validity of the findings

For clarification:

In Figure 2, A, please make it clear that the arrows are directed at the SA-3-gal-stained cells. Also, it is quite confusing to see the given stains on 0 ug/ml or control samples. Please clarify this part.

Please check as well: yH2A.X on Figure 3. 40 ug/ml seems to have more expression than 80 ug/ml. This part can be confusing.

Additional comments

Generally, the paper was well-thought and organized. The way it was written is also orderly. Minor revisions are needed.

·

Basic reporting

There are minor corrections in the following line numbers: 68, 84, 85, 117, 140.

Experimental design

No comment.

Validity of the findings

No comment

Additional comments

Altogether the team has done a tremendous job. Congratulations and best wishes!!

Reviewer 3 ·

Basic reporting

1) Please place the citation first, then put the dot to end sentence. Please make necessary changes in whole manuscript and please revise the in text citation format acording to the journal's guidelines.

2) You can use "healthy" or any other appropriate term instead of "normal". Please revise in whole manuscript.

3) Please use "negative control" instead of "normal control". Please revise in whole manuscript.

4)There are some parts which could be used in a different section.

4) The manuscript lacks a conclusions section. Please make a conclusions section.

5) Please improve the introduction section. There's no mention of arecoline in this section. We learn about the betel nuts in materials and methods section which its consumption is linked with OSF by other researchers, etc.

6) "2.5 Induction of HOKs senescence by arecoline" section is repeating the section before. Please avoid repetation. If there's nothing more to tell, you can delete this section.

7)There are a few places, needing citations.

Please see the uploaded pdf file for my further inquiries.

Experimental design

1) Methodologies for fixation, tissue processing, embedding, tissue sectioning should be added. Determination of the used doses of aceroline should be mentioned as well.

2) In the IHC section:
-Which blocking serum was used?
-Did authors performed an antigen retrieaval?
-Which secondary antibody was used?
-Which chromogen was used?
-How these areas of interest were selected?
*Please place a citation for quantification method.

3) Please briefly describe the Sudan Black staining.

4)Instead of giving a citation, please describe the cell culture. Indicating the growth medium and culture conditions are important to see.

5) For qPCR section:
-Which RNA quantification method was used?
-Please mention the use of GAPDH as the housekeeping gene.
-Please mention which group was used to calculate ∆∆Ct values before the fold change in expression value.
-Please add Tm values for each PCR primer in Table 1.

6) Authors mention that the manufacturer provided a standard curve for their ELISA. Could you please share this standard curve provided by the manufacturer? How did manufacturer knew your blanks and how did manufacturer expect to see same readings in standard solutions?

7) There's something bothering me with these results. We see that 320 ug/ml dose of arecoline is resulting with significantly lower cell proliferation rates and we see significantly different results only for this dose. In addition to that, despite having same bar sizes in IF images, we see much less nuclei in the images compared to other doses, which indicates that there are much less cells in 320 ug/ml aceroline group. It could be due to either cellular senescence, or cell death which should be clarified. In the report of Jheng et al(a), they described 100 ug/ml of aceroline as cytotoxic. Also, Chiang et al (b) indicates that 200 ug/ml arecoline causes much higher cytotoxicity in human gingival fibroblasts. Have you made any cytotoxicity assays prior to this study with aceroline on these cells in these doses? Have you performed any analyses for determining IC50 value for aceroline?

a) Jeng, J. H., et al. "Genotoxic and non-genotoxic effects of betel quid ingredients on oral mucosal fibroblasts in vitro." Journal of Dental Research 73.5 (1994): 1043-1049.

b)Chiang, Shang-Lun, et al. "Characterization of arecoline-induced effects on cytotoxicity in normal human gingival fibroblasts by global gene expression profiling." Toxicological Sciences 100.1 (2007): 66-74.

Validity of the findings

1) I believe the study given below and/or the studies used in this review should be used in your discussion. There are parts indicating the senescence of myofibroblasts in OSF as well, which could be useful in your discussion
(https://onlinelibrary.wiley.com/doi/abs/10.1002/hed.26805)
Sharma, M., Hunter, K. D., Fonseca, F. P., & Radhakrishnan, R. (2021). Emerging role of cellular senescence in the pathogenesis of oral submucous fibrosis and its malignant transformation. Head & Neck, 43(10), 3153-3164.

2)In which concentrations we can say that arecoline had successfully resulted in cellular senescence of HOKs? Only significant results were obtained from 320 ug/ml. Was it only this dose, or are there another doses which we can say resulted in senescence?

3) Regarding the TGF ß1, there are a few useful articles which could be used in the discussion
-This article links TGF-ß1 and oral cancer and fibroblast senescence:
Tan, May Leng, et al. "Autophagy is deregulated in cancer-associated fibroblasts from oral cancer and is stimulated during the induction of fibroblast senescence by TGF-β1." Scientific reports 11.1 (2021): 1-14.
APA( https://www.nature.com/articles/s41598-020-79789-8 )
-this article also mentions TGF-ß and senescence of cancer associated sfibroblasts in oral cancer
Hassona, Yazan, et al. "Progression of genotype-specific oral cancer leads to senescence of cancer-associated fibroblasts and is mediated by oxidative stress and TGF-β." Carcinogenesis 34.6 (2013): 1286-1295.
( https://academic.oup.com/carcin/article/34/6/1286/2463038 )

There are a few more studies. Please try to catch more linkages with prior studies and improve the discussion section for all your results.

Additional comments

My further inquiries could be seen in the uploaded pdf file.

Annotated reviews are not available for download in order to protect the identity of reviewers who chose to remain anonymous.

---

## Round 0.2 · Minor Revisions

Based on Reviewer 3, the article needs minor revision.

Reviewer 1 ·

Basic reporting

Thank you for including the significance of TGF-β1 on OSF in the Introduction.

Please double-check the grammar and other sentence constructions.

Please write the complete wordings (ECM, EMT...) since they are crucial points in Cancer Biology.

Experimental design

No further comments. Thank you.

Validity of the findings

The yH2A.X in Figure 3 remains ambiguous to me as the image is not as clear at a certain scale.

The rest are acceptable to me.

·

Basic reporting

No comments

Experimental design

No comments

Validity of the findings

Has already checked and has no comments

Reviewer 3 ·

Basic reporting

1) Bettel nut and aceroline are two of the focus points of the study. Yet all we learn from the introduction is "Betel nut is one of the critical pathogenic factors for OSF, and its main component arecoline has been confirmed to be closely linked to OSF progression." Firstly, this statement needs citations. Secondly, this information is not a satisfactory representation. Further elaborations and linkages with the problem statement are necessary.

2)Line 85: "TGF-β1 is a multifunctional cytokine that plays a role in numerous physiological and pathological processes."
Needs citation

Experimental design

1) In their rebuttal, authors stated that "IC50 values for arecoline on HOKs were 200 μg/ml (24h) and 160 μg/ml (48h) respectively". Yet, readers still do not see anything about it. It should be mentioned in the manuscript for future studies.

Validity of the findings

1) Authors stated that "The TGF-β1 concentration of each sample was determined by a standard curve provided from the manufacturer’s protocol." Did authors made their own standard curve or they used the standard curve presented in the manufacturers datasheet?

---

## Round 0.3 · accepted · Accept

The manuscript can be accepted.